# Resveratrol Ameliorates Vancomycin-Induced Testicular Dysfunction in Male Rats

**DOI:** 10.3390/medicina59030486

**Published:** 2023-03-01

**Authors:** Fahad S. Alshehri

**Affiliations:** Department of Pharmacology and Toxicology, College of Pharmacy, Umm Al-Qura University, Makkah 24382, Saudi Arabia; fsshehri@uqu.edu.sa

**Keywords:** vancomycin, resveratrol, testosterone, sperm motility, sperm counts

## Abstract

*Background and Objectives*: Numerous studies have indicated that antibiotics may adversely affect testicular and sperm function. As an alternative to penicillin, vancomycin is a glycopeptide antibiotic developed to treat resistant strains of Staphylococcus aureus. A few studies have suggested that vancomycin could cause testicular toxicity and apoptosis. Vancomycin, however, has not been investigated in terms of its mechanism of causing testicular toxicity. *Materials and Methods:* An experiment was conducted to investigate the effects of resveratrol (20 mg/kg, oral gavage) against vancomycin (200 mg/kg, i.p.) on the testicular function of Wistar rats for one week (7 days). There were three subgroups of animals. First, saline (i.p.) was administered to the control group. Then, in the second group, vancomycin was administered. Finally, vancomycin and resveratrol were administered in combination in the third group. *Results:* After seven days of vancomycin treatment, testosterone levels, sperm counts, and sperm motility were significantly reduced, but resveratrol attenuated the effects of vancomycin and restored the testosterone levels, sperm counts, and sperm motility to normal. In the presence of resveratrol, the vancomycin effects were attenuated, and the luteinizing hormone and follicular hormone levels were normalized after seven days of treatment with vancomycin. Histologically, vancomycin administration for seven days caused damage to testicular tissues and reduced the thickness of the basal lamina. However, the resveratrol administration with vancomycin prevented vancomycin’s toxic effects on testicular tissue. *Conclusion:* Resveratrol showed potential protective effects against vancomycin-induced testicular toxicity in Wistar rats.

## 1. Introduction

There has been a growing body of research indicating that antibiotics may adversely affect sperm and testicular function [1,2,3,4]. Generally, vancomycin is used for infections caused by Methicillin-resistant Staphylococcus aureus (MRSA) and for patients allergic to penicillin or cephalosporins [5,6]. It has been reported that higher vancomycin trough concentrations should be achieved because MRSA-related infections require higher minimum inhibitory concentrations due to the difficulty in penetrating vulnerable sites such as the lungs, brain, and bone [7,8,9]. However, vancomycin treatment is associated with kidney- and liver-related side effects [10,11,12,13], making it unsuitable for patients with impaired renal and hepatic function [10,11]. 

There has been significant evidence that various antibiotics can impair the motility of sperm, reduce the weight of the reproductive organs, and cause apoptosis in the testes, eventually leading to testicular failure [12]. A large body of research has shown that the use of antibiotics can disrupt the normal functioning of the male reproductive system by interfering with the hormones that regulate spermatogenesis and sperm motility. Additionally, antibiotics can damage sperm cells, leading to reduced sperm count and motility, which can eventually cause infertility [2,13,14,15]. Several recent studies have reported that vancomycin may adversely affect the testes and cause apoptosis in some cases. [16,17]. It is still unclear how vancomycin causes testicular toxicity. It has been shown that vancomycin increases oxidative stress, such as elevated lipid peroxide levels, and reduces antioxidant enzymes, such as glutathione (GSH) and mitochondrial damage [18,19]. Furthermore, vancomycin stimulates reactive oxygen species (ROS) and inhibits DNA synthesis [20,21]. The presence of ROS is physiologically essential in semen; however, higher levels of ROS production exceed the natural sperm antioxidant ability to prevent ROS damage [22]. Thus, higher levels of free radicals can reduce spermiogenesis resulting in loss of motility and DNA damage in the sperm nucleus [23]. 

Resveratrol (3,4′,5-trihydroxy-trans-stilbene) is a natural polyphenolic compound found in vegetables such as grapes, berries, and peanuts. Resveratrol has continuously been reported to have a growing number of benefits. For example, cumulative reports have suggested that resveratrol has potential benefits as an anti-inflammatory, antidiabetic, anticancer, and protective effect against cardiovascular disease. Moreover, resveratrol could improve stress resistance, extend human lifespans, and prevent the progression of many illnesses, including cancer, cardiovascular disease, and ischemic injuries [24,25,26]. As a result, resveratrol’s antioxidant properties have effectively protected cells from hydrogen-peroxide-induced oxidative stress and UV-irradiation-induced cell death after pretreatment with resveratrol [27,28,29]. Furthermore, in pharmaceutical products, resveratrol can delay the oxidation of lipids, reduce the toxic byproducts of oxidation, and extend the shelf life while maintaining the nutritional quality [30,31]. The structure of resveratrol is similar to that of estradiol, which suggests that it may play a similar role in the testes [32]. In humans and domestic animals, resveratrol has been shown to improve sperm quality [33]. This appears to be possible as a result of its ability to pass through the blood–testis barrier and impart its protective properties to the testes [34]. The use of resveratrol to treat infertility in vivo has been found to be effective. Resveratrol has been found to be effective in treating men afflicted with dyszoospermia because it ameliorated the effect induced by 2,5-hexanedione on spermatogenesis [32]. It has been shown that oral administration of resveratrol and coenzyme Q10 protects against radiation-induced spermatogenesis injuries, suggesting that the combination may be beneficial for promoting male fertility [35]. Even though considerable research has been conducted, it is unclear what role resveratrol plays in male reproductive function.

Therefore, many attempts have been implemented to reduce antibiotics’ effects on spermatogenesis [36,37]. It is well known that resveratrol is an antioxidant that can scavenge ROS, preventing the damage of cells in tissues. There is promising evidence that antioxidants effectively preserve spermatogenesis in an animal model and treat male factor infertility. Several studies have shown that vancomycin can cause testicular atrophy and impaired sperm quality in animals and humans. However, limited information about vancomycin’s effect on men with reproductive disorders is available. This study aimed to determine whether high doses of vancomycin induce testicular or spermatotoxicity in rats and to investigate whether resveratrol might have a modulatory effect on the development of testicular damage induced by high doses of vancomycin.

## 2. Material and Methods

### 2.1. Drugs

During the study, resveratrol was obtained from ProHealth in the USA (B094XH3W98), mixed with saline (0.9% NaCl) as suspension, and given to animals immediately, and vancomycin was obtained from Medis in Tunisia (AMM12/96860), dissolved in saline (0.9% NaCl). The dose of resveratrol was selected based on several studies that used 20 mg/kg to show its antioxidant and anti-inflammatory effects [38,39,40]; the vancomycin dose was selected based on its ability to produce toxic effects on several organs, including testicular tissues [41,42,43].

### 2.2. Animals

Twenty-one adult male Wistar rats weighing 160–200 g (7 weeks old) were used in the study. The animals were housed in plastic cages in a humidity-controlled room under a 12 h light/12 h dark schedule for the experiment. Daily monitoring was performed to ensure the wellbeing of the animals, who had access to food and water ad libitum. All animals were obtained from the King Abdulaziz University animal house. 

### 2.3. Experimental Design

The experiment was conducted for eight days. The rats were divided into three groups: (1) control group (*n* = 7), rats injected with saline (i.p) for seven days; (2) vancomycin group (*n* = 7), rats injected with vancomycin (200 mg i.p) for seven days; (3) vancomycin + resveratrol group (*n* = 7), rats injected with vancomycin (200 mg i.p) and resveratrol (20 mg/kg, oral gavage) for seven days (Table 1). On day eight, all rats were euthanized using CO_2_; then, serum and tissue samples were collected. The blood samples were collected from their retroorbital plexus and tail vain. For the separation of serum from plasma, a blood sample was collected and centrifuged at 3000× *g* for 15 min at 4 °C. For testing, all samples were centrifuged and analyzed immediately.

### 2.4. Determination of Testosterone, Follicle Stimulating Hormone (FSH), and Luteinizing Hormone (LH) by ELISA

Serum samples were collected on day 8. Each sample was centrifuged, and each animal’s serum was analyzed separately. The ELISA technique was used to determine the serum levels of testosterone, FSH, and LH according to the manufacturer’s protocol (MyBioSource, Inc San Diego, CA 92195-3308, USA).

### 2.5. Determination of the Sperm Motility and Counts

The male rats were euthanized by CO_2_ inhalation. The sperm samples were diluted with physiological solution (10 μL), pipetted with TL-HEPES solution containing 3 mg/mL bovine serum albumin, and then used to buffer HEPES-buffered Tyrode lactate (TL-HEPES) solution. The cauda epididymis was cut at several points at 37 °C to allow the sperm to flow out. The sperm motility percentage and sperm count (Cells/mm^3^) were obtained using computer-assisted semen analysis to measure the sperm motility and forward motility with the SpermVision™ CASA System (MiniTub, Tiefenbach, Germany). Following Zemjanis’ method, the sperm motility of rats was measured within 2–4 min of sacrifice [44].

### 2.6. Histological Examination

The testicular tissues were used for histopathological assessment and prepared in 10% formalin solution for two days. The tissue was embedded in paraffin blocks and stained with Hematoxylin and Eosin (H&E). The thickness of the slices was between 3 and 5 mm. An OMAX 3 MP Digital Compound Microscope was used to observe the stained sections at various magnifications and to take photographic micrographs. Under a light microscope at ×400, 20 seminiferous tubules from each animal section were evaluated for histomorphometric changes. In addition, histopathological changes in the testes were examined.

### 2.7. Statistical Analysis

The serum testosterone, follicle-stimulating hormone, luteinizing hormone, and sperm motility were analyzed using a one-way analysis of variance (ANOVA) followed by Tukey’s post hoc tests. The data were analyzed using Prism version 9.4.1 (*p*-value < 0.05).

## 3. Results

### 3.1. Body Weight

There was no significant difference in the body weight of all the treated groups compared to the controls on day 1. However, repeated measure two-way ANOVA analysis showed a significant main effect in days, as shown in the ANOVA table [F (1, 6) = 178.5, *p* < 0.0001], treatment [F (2, 12) = 4.026, *p* = 0.0459], the days x treatment [F (2, 12) = 126.1, *p* < 0.0001] (Figure 1). In addition, multiple comparison tests using the Tukey post hoc test revealed a significant reduction in body weight in the vancomycin group on day 8 compared to the vancomycin group on day 1 (*p* < 0.0001). Moreover, there was a reduction in body weight in the vancomycin group compared to the control group on day 8 (*p* < 0.0001). However, there was an increase in the body weight of the vancomycin + resveratrol group compared to the vancomycin group (*p* < 0.0001).

### 3.2. Testosterone, Follicle Stimulating Hormone, and Luteinizing Hormone in the Blood Levels

The testosterone level in the blood was determined on day 8 of the experiment. One-way ANOVA revealed significant changes in the level of testosterone (ng/dl) between the treatment groups, as shown in the ANOVA table [F (2, 18) = 17.02, *p* < 0.0001, Figure 2]. Further analysis using Tukey’s multiple comparison tests showed a significant reduction in the testosterone levels in the vancomycin group compared to the control group (*p* = 0.0002) and the vancomycin + resveratrol group (*p* = 0.0003). However, no significant changes were found between the control group and the vancomycin + resveratrol group (*p* = 0.9977).

The FSH level in the blood was determined on day 8 of the experiment. One-way ANOVA revealed significant changes in the level of FSH (mIU/mL) between the treatment groups, as shown in the ANOVA table [F (2, 18) = 41.43, *p* < 0.0001, Figure 3]. Further analysis using Tukey’s multiple comparison tests presented a significant increase in the FSH levels in the vancomycin group compared to the control group (*p* < 0.0001) and the vancomycin + resveratrol group (*p* < 0.0001). However, no significant changes were found between the control group and the vancomycin + resveratrol group (*p* = 0.8106).

The LH level in the blood was determined on day 8 of the experiment. One-way ANOVA revealed significant changes in the level of LH (mIU/mL) between the treatment groups, as shown in the ANOVA table [F (2, 18) = 50.58, *p* < 0.0001, Figure 4]. Further analysis using Tukey’s multiple comparison tests showed a significant increase in the LH levels in the vancomycin group compared to the control group (*p* < 0.0001) and the vancomycin + resveratrol group (*p* < 0.0001). However, no significant changes were found between the control group and the vancomycin + resveratrol group (*p* = 0.8745).

### 3.3. Sperm Motility and Counts

The sperm motility was determined on day 8 of the experiment. One-way ANOVA revealed significant changes in the sperm motility between the treatment groups, as shown in the ANOVA table [F (2, 18) = 21.70, *p* < 0.0001, Figure 5]. Further analysis using Tukey’s multiple comparison tests showed a significant reduction in the sperm motility in the vancomycin group compared to the control group (*p* < 0.001) and the vancomycin + resveratrol group (*p* < 0.0001). However, no significant changes were found between the control group and the vancomycin + resveratrol group (*p* = 0.9314).

The sperm counts were determined on day 8 of the experiment. One-way ANOVA revealed significant changes in the sperm counts (Cells/mm^3^) between the treatment groups, as shown in the ANOVA table [F (2, 18) = 29.55, *p* < 0.0001, Figure 6]. Further analysis using Tukey’s multiple comparison tests showed a significant reduction in the sperm motility in the vancomycin group compared to the control group (*p* < 0.0001) and the vancomycin + resveratrol group (*p* = 0.0001). However, no significant changes were found between the control group and the vancomycin + resveratrol group (*p* = 0.1148).

### 3.4. Histological Examinations of the Testicles

The histology of the testicles was not significantly different between the rats given daily saline alone and the controls (Figure 7A–C). Hence, it was found that normal spermatogenesis had taken place, that the Sertoli cells had been preserved well, and that the tubular basement membrane had been clearly defined. Furthermore, the interstitial space between the tubules and the Leydig cells also appeared intact. However, the vancomycin-treated group showed a significant difference in the histology of the testes, where the seminiferous tubules were observed to be swallowed up completely. The tubular basement membranes of the seminiferous tubules were identified in other areas of the section. While most germ cells, including highly differentiated germ cells and deformed sperm, were degenerating, a small percentage were flourishing. It is also important to note that the ground substance within the interstitium partially disappeared and was replaced by fibroblasts and inflammatory cells. There was an improvement in these toxic effects in the group treated with vancomycin and resveratrol.

## 4. Discussion

A growing concern has been raised over the possibility that antibiotics may adversely affect human fertility [3,13,45]. Vancomycin is considered one of the most common antibiotics used globally for treating severe Gram-positive infections caused by meticillin-resistant S aureus (MRSA) [5,46]. Nevertheless, vancomycin has long been recognized as one of the most commonly encountered drugs that induces nephrotoxicity and hepatotoxicity. In addition to its toxic effects, vancomycin can also cause testicular toxicity [43]. Hence, the present study was conducted to investigate the protective effects of resveratrol against the toxic effects of vancomycin on the testicular functions of male Wistar rats, through analysis of the histopathological and biochemical profiles.

The present study demonstrated the toxicological effects of vancomycin in male Wistar rats. There was a significant decrease in the serum testosterone levels after administering vancomycin. Low intratesticular testosterone concentrations may result in germ cell degeneration due to vancomycin exposure. It has been suggested that the testosterone level in the testes is essential for spermatogenesis and maintaining the seminiferous tubules’ structural morphology and physiology [47,48,49]. Vancomycin administration was associated with the degeneration of germ cells, including highly differentiated germ cells and deformed sperm. Moreover, changes in the testes’ morphological characteristics in the vancomycin treatment groups were observed. Epithelium, tubular shrinkage, and atrophy were manifestations of these changes.

In addition, several studies have demonstrated that the sperm count and motility are the most valuable indicators of male fertility. Research shows that the sperm count and motility are positively associated with pregnancy rates. Based on the findings of our study, the vancomycin-induced structural damage to rat testicular tissues resulted in a severe reduction in the sperm count and motility. Furthermore, it has been suggested that administering vancomycin can lead to various biochemical malfunctions [50,51]. Unfortunately, there is a lack of understanding the mechanisms through which vancomycin produces these effects. There is, however, evidence that reactive oxygen species (ROS) are involved. In fact, the rats exposed to vancomycin showed an elevation in oxidative stress caused by a reduction in antioxidant enzymes, such as the glutathione levels, coupled with an increase in the lipid peroxide levels, which resulted in oxidative stress in the animals [16,52]. Other antibiotics have shown similar patterns regarding testicular dysfunction. In vivo and in vitro, it has been well established that gentamicin is capable of causing ROS formation and oxidative damage. In the testes of rats treated with gentamicin, a similar reduction in enzymatic and nonenzymatic antioxidant activity was observed, along with increased lactoperoxidase levels [53]. As a result of the treatment with gentamicin, the MDA concentrations increased, and the GSH levels decreased. Consequently, the oxidative stress caused by the gentamicin treatment could be linked to increased lactoperoxidase levels and decreased antioxidant activity and GSH levels in the testes [54]. Therefore, it is essential to monitor antibiotic use to minimize the potential risks.

Moreover, there was a significant increase in the serum LH and FSH levels after administering vancomycin. The LH level was also significantly higher in the treatment group, indicating that the hypophyseal–pituitary axis was affected [55]. In fact, male interstitial cells produce testosterone in response to LH stimulation [56]. Additionally, the FSH level was significantly higher than that of the controls. It has been reported that there is a direct interaction between FSH and Sertoli cells; therefore, FSH binds to its receptor and stimulates its signaling pathway, which leads to Sertoli cell differentiation [57]. It is possible that an elevated level of FSH indicates abnormal spermatogenesis and may indicate testicular failure [58]. Conversely, a low level of testosterone and raised FSH and LH levels have been associated with insufficient sperm production by the testicles [59]. As a result of these changes, this study suggests that using vancomycin in high doses is likely to lead to infertility.

Known for its antioxidant properties, resveratrol is a natural polyphenolic compound present in vegetables such as grapes, berries, and peanuts [60]. In recent years, researchers have extensively studied the antiaging properties of resveratrol and its potential to help prevent aging-related diseases, such as Alzheimer’s and diabetes [61,62]. Furthermore, resveratrol may play a role in preventing heart disease and stroke [63] and improving cognitive function [64]. According to cumulative reports, resveratrol has potential anti-inflammatory and protective effects against cancer [65]. Moreover, resveratrol could improve stress resistance, extend human lifespans, and prevent the progression of many illnesses [66]. Further, it has recently been shown that in vivo treatment with resveratrol prevents oxidative stress in the testes of hyperthyroid rats and rats treated with a chemotherapy drug [33,67]. Although resveratrol appears to have an antioxidant effect on male reproduction, its exact mechanism of action is unknown. As part of this study, resveratrol was examined for its protective effects against damage caused by vancomycin on rats’ spermatozoa. The protective effects of resveratrol on the glutathione levels have been reported, particularly those conducted in a testicular ischemia model. Furthermore, resveratrol stimulates spermatogenesis and testicular regeneration in adults [32]. In addition, rats treated with resveratrol produced more spermatozoa [68]. A similar increase in spermatozoa production was observed in the resveratrol groups compared to the vancomycin groups. In the present study, we observed that resveratrol prevented the sperm reduction caused by vancomycin, suggesting that it has antioxidant properties. The oxidative stress produced by vancomycin, including hydroxyl radicals, can be scavenged effectively by resveratrol [69]. However, several possible molecular mechanisms could explain the effects of resveratrol. It has been reported that resveratrol maintains the integrity of mitochondrial membranes and provides sufficient energy to spermatogonial stem cells through modulating the SIRT1 protein and deacetylating FOXO1 in vitro [70]. Moreover, a study conducted in rats showed that resveratrol increased the expression of sirtuin-1, neuronal nitric oxide synthase (nNOS), decreased the rate of cell death, and stimulated the differentiation of germ cells in rats [35,71,72,73]. However, the mechanisms of action of resveratrol remain unclear, even though it protects spermatozoa against oxidative stress.

## 5. Conclusions

In conclusion, it is becoming increasingly apparent that antibiotics may adversely affect human fertility. Vancomycin was investigated in this study to understand the possible protective effects of resveratrol on vancomycin-induced testicular toxicity. This study investigated the protective effects of resveratrol against vancomycin’s toxic effects on the testicular functions of adult male Wistar rats through analyzing the histopathological and biochemical profiles.

## Figures and Tables

**Figure 1 medicina-59-00486-f001:**
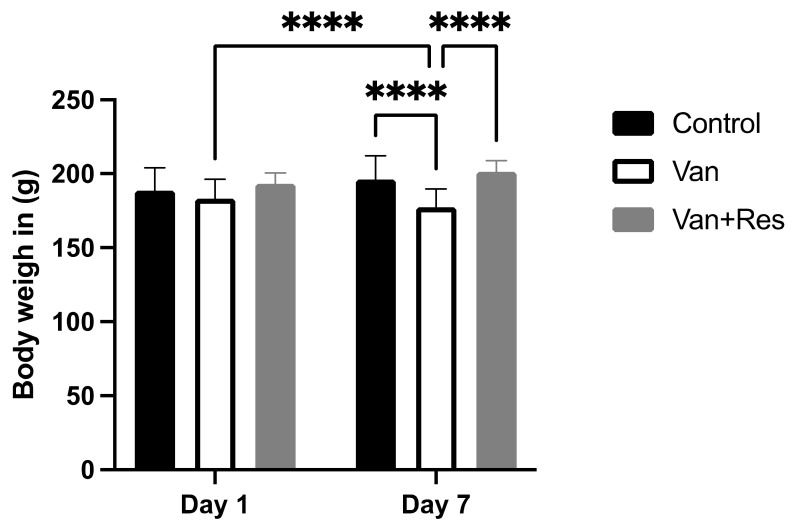
The effect of the resveratrol and vancomycin administration on the animals’ body weights during the experiment. **** *p* < 0.0001.

**Figure 2 medicina-59-00486-f002:**
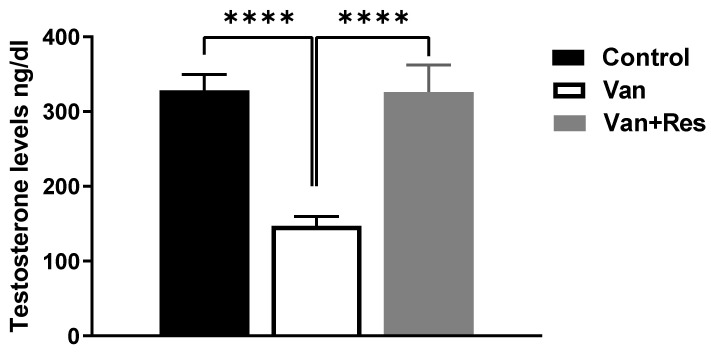
The effect of the resveratrol and vancomycin administration on the testosterone during the experiment. **** *p* < 0.0001.

**Figure 3 medicina-59-00486-f003:**
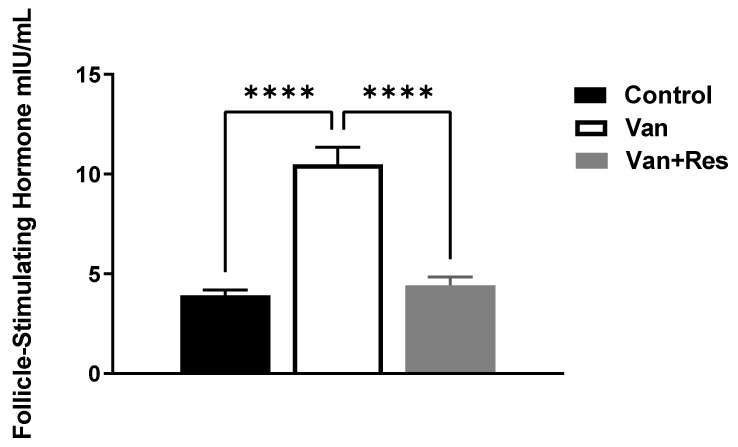
The effect of the resveratrol and vancomycin administration on the follicle-stimulating hormone during the experiment. **** *p* < 0.0001.

**Figure 4 medicina-59-00486-f004:**
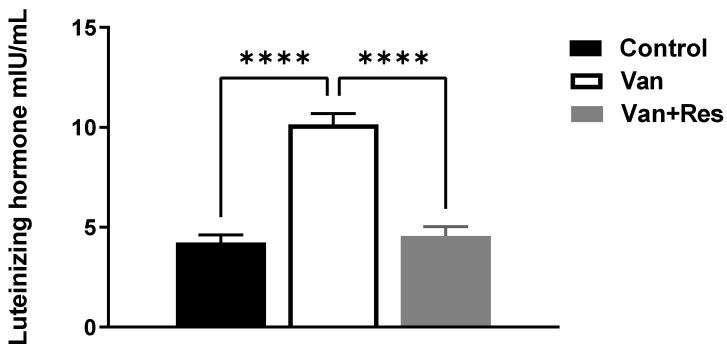
The effect of the resveratrol and vancomycin administration on the luteinizing hormone during the experiment. **** *p* < 0.0001.

**Figure 5 medicina-59-00486-f005:**
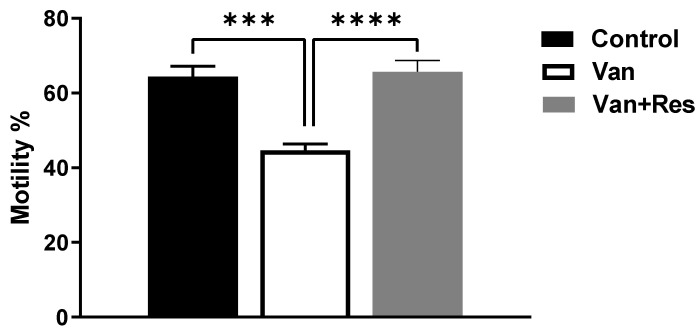
The effect of the resveratrol and vancomycin administration on the motility during the experiment. *** *p* < 0.001, **** *p* < 0.0001.

**Figure 6 medicina-59-00486-f006:**
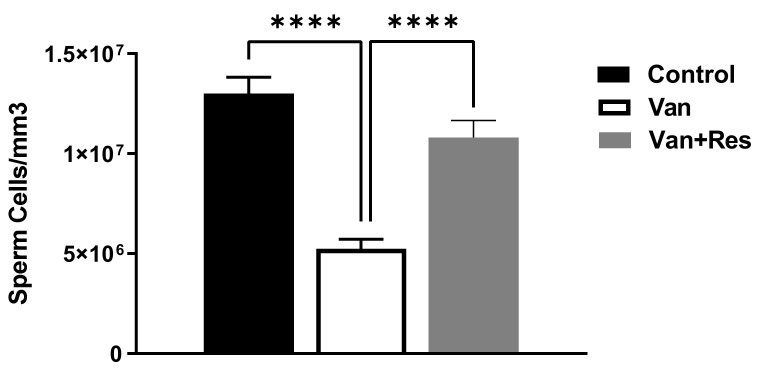
The effect of the resveratrol and vancomycin administration on the sperm during the experiment. **** *p* < 0.0001.

**Figure 7 medicina-59-00486-f007:**
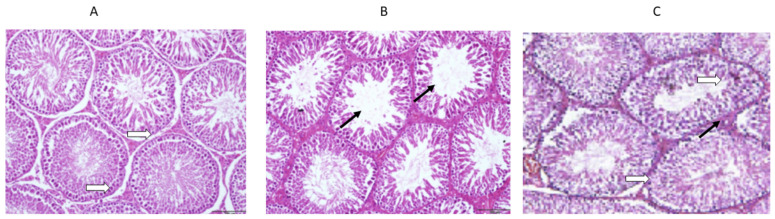
(**A**) The effect of the normal saline administration during the experiment on the histological sections of rat testes was a normal morphology with a normal thickness of the basal lamina (white arrows). (**B**) The effect of the vancomycin administration on the histological sections of rat testes was thinning and splitting of the basal lamina, with atrophy and widely separated seminiferous tubules (black arrows). (**C**) The effect of the resveratrol and vancomycin administration on the histological sections of rat testes was a normal thickness of the basal lamina (white arrows) and normal seminiferous tubules (black arrows).

**Table 1 medicina-59-00486-t001:** Experimental groups for the resveratrol and vancomycin administration.

Group	Treatment
Control	Normal saline
Vancomycin	200 mg/kg, i.p for seven days
Vancomycin and resveratrol	Vancomycin 200 mg/kg (i.p) and 20 mg/kg resveratrol (oral gavage), for seven days

## Data Availability

The original data will be avalaible upon request from the author.

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
