# Peer review of "Resveratrol Ameliorates Vancomycin-Induced Testicular Dysfunction in Male Rats"

_medicina, 2023, doi:10.3390/medicina59030486_

Round 1

Reviewer 1 Report

In this study, the prevention of testicular injury due to vancomycin with resveratrol was investigated. Regardless of the agent, the prevention of testicular toxicity is still a subject that needs a lot of research. In this respect, it is an important study that will guide clinical applications. However, a more detailed study design could have been made.

1.     In material and methods, the catalog numbers of Vancomycin and Resveratrol should be specified. In particular, how Resveratrol is prepared should be explained in detail. If it is in powder form, it should be stated how the dissolution and dilution method was and how the dose was adjusted. For example, whether dimethyl sulfoxide (DMSO) or ethylalcohol or ddH2O was used in these processes should be explained. Since resveratrol is poorly water soluble, carboxymethylcellulose could be used.

2.     The effect of resveratrol is tissue-specific and dose-dependent. For this reason, it should be stated according to which source the dose determination was made.

3.     In Table 1, the route of administration of resveratrol appears as if it were intraperitoneal. However, it is made with oral gavage. This should also be specified.

4.     The way of collecting blood samples from animals should be specified. Were the hormonal kits used specific to rats? Were the hormone measurements done separately for each animal, or were they frozen and used later for evaluation together?

5.     Which staining method was used in the histopathological examination of testicular tissues should be explained. It should be also stated in how many areas the histological examinations were made and whether morphometric measurements were used.

6.     Which criterion was used while evaluating sperm motility, should be written. It can be forward-fast, forward-slow, or just forward.

7.     In statistical expressions, the letter "p" should be changed to italicized "p" so that it is the same throughout the text.

8.     When comparing the significance of the results in the tables, the meaning of the "****" sign should be stated in the legend.

9.     Line 118: The meanings of the expressions F(1, 6) and F (2, 12) in the results section should be written clearly. The distribution of these findings according to the groups should be expressed in a more understandable way.

10.  In Discussion: Line 205-207: It is not clear what "meticillin" means here. The sentence needs to be revised.

11.  Line 223-224: Appropriate reference should be given.

12.  Line 235: Not “There”, but “there”.

13.  Line 239, 242 and 251: Appropriate references should be given.

14.  The protective or restorative effects of resveratrol on testicular functions cannot be explained by oxidative stress alone. In addition, numerous resveratrol-related signaling pathways may be reactivated and thereby regulate many cell activities including intracellular redox cycle and proliferation, and cell death. In addition, the effects of resveratrol on epithelial-interstitial transformation process and cell proliferation have also been demonstrated. Even anti-apoptotic effects through tumor suppressor gene induction have been suggested. A comprehensive literature review should be carried out and the mechanisms of action of resveratrol should be detailed in the discussion section (Example sources: doi: 10.18632/oncotarget.12990, doi: 10.1093/jn/135.4.757 (Ref 41), doi: 10.3390/molecules25194554).

Author Response

Reviwer 1

In this study, the prevention of testicular injury due to vancomycin with resveratrol was investigated. Regardless of the agent, the prevention of testicular toxicity is still a subject that needs a lot of research. In this respect, it is an important study that will guide clinical applications. However, a more detailed study design could have been made.

  1. In material and methods, the catalog numbers of Vancomycin and Resveratrol should be specified. In particular, how Resveratrol is prepared should be explained in detail. If it is in powder form, it should be stated how the dissolution and dilution method was and how the dose was adjusted. For example, whether dimethyl sulfoxide (DMSO) or ethylalcohol or ddH2O was used in these processes should be explained. Since resveratrol is poorly water soluble, carboxymethylcellulose could be used.

I would like to thank the reviewer, we added further details in the method section under drugs

  1. The effect of resveratrol is tissue-specific and dose-dependent. For this reason, it should be stated according to which source the dose determination was made.

I would like to thank the reviewer, we added further details in the method section under drugs

  1. In Table 1, the route of administration of resveratrol appears as if it were intraperitoneal. However, it is made with oral gavage. This should also be specified.

I would like to thank the reviewer. The table was updated as requested

  1. The way of collecting blood samples from animals should be specified. Were the hormonal kits used specific to rats? Were the hormone measurements done separately for each animal, or were they frozen and used later for evaluation together?

I would like to thank the reviewer, we added further details in the method section.

  1. Which staining method was used in the histopathological examination of testicular tissues should be explained. It should be also stated in how many areas the histological examinations were made and whether morphometric measurements were used.

I would like to thank the reviewer, we added further details in the method section.

  1. Which criterion was used while evaluating sperm motility, should be written. It can be forward-fast, forward-slow, or just forward.

I would like to thank the reviewer, we added further details in the method section, and the reference was added.

  1. In statistical expressions, the letter "p" should be changed to italicized "p" so that it is the same throughout the text.

I would like to thank the reviewer, the “p” was updated as requested through the manuscript.

  1. When comparing the significance of the results in the tables, the meaning of the "****" sign should be stated in the legend.

I would like to thank the reviewer, the “****” was updated as requested through the legend.

  1. Line 118: The meanings of the expressions F(1, 6) and F (2, 12) in the results section should be written clearly. The distribution of these findings according to the groups should be expressed in a more understandable way.

I would like to thank the reviewer, the ANOVA table was written to explain it more as requested through the manuscript.

  1. In Discussion: Line 205-207: It is not clear what "meticillin" means here. The sentence needs to be revised.

  I would like to thank the reviewer, it was a typo error. 

  1. Line 223-224: Appropriate reference should be given.

  I would like to thank the reviewer, the reference was added

  1. Line 235: Not “There”, but “there”.

   I would like to thank the reviewer, it was a typo error. 

  1. Line 239, 242 and 251: Appropriate references should be given.

  I would like to thank the reviewer, the references were added

  1. The protective or restorative effects of resveratrol on testicular functions cannot be explained by oxidative stress alone. In addition, numerous resveratrol-related signaling pathways may be reactivated and thereby regulate many cell activities including intracellular redox cycle and proliferation, and cell death. In addition, the effects of resveratrol on epithelial-interstitial transformation process and cell proliferation have also been demonstrated. Even anti-apoptotic effects through tumor suppressor gene induction have been suggested. A comprehensive literature review should be carried out and the mechanisms of action of resveratrol should be detailed in the discussion section (Example sources: doi: 10.18632/oncotarget.12990, doi: 10.1093/jn/135.4.757 (Ref 41), doi: 10.3390/molecules25194554).

I would like to thank the reviewer for suggesting these references; a more detailed discussion about the possible mechanisms of resveratrol was added.

Reviewer 2 Report

In the submitted manuscript (medicina-2232959), the author examined the effect of a short administration of the antibiotic vancomycin on testicular function in rats, as well as the protective effect of the antioxidant resveratrol on worsened monitored parameters, such as the level of testosterone, follicle-stimulating hormone, luteinizing hormone in the blood, sperm count and mobility. Morphological changes in the testicles as a consequence of the treatment were also monitored. Under the applied conditions of the experiment, the author reports dramatic changes in the monitored parameters in the animals treated with the antibiotic, especially those in the blood, which are all normalized (i.e., they corresponded to the controls) if the antibiotic is added with an antioxidant.

My overall evaluation of the existing text is not favorable. Apart from the idea of the entire study, which is excellent, everything else, starting with the design of the study, the lack of details of the performed experiments, the description and commenting of the obtained findings, and their placement in the context of the literature in this field, is unfortunately not at the required level.

The paper's technical quality should be much better, avoiding unclear sentences, countless typing and editing errors, material mistakes, and unnecessary repetitions (for example, it is enough to say once in the summary of the work how long the treatments lasted), etc.

The most critical essential remarks are the following:

1. Duration of the treatment: why did the author decide that it only lasts seven days? Vancomycin is an antibiotic used in humans for severe pathophysiological conditions for longer than one week.

2. Why did the author apply a daily dose of antibiotics of a large 200 mg (i.p.) to a body weight of only 200 g? How old were the experimental animals anyway? Adult ones are usually heavier (more than 300 g after three months of life). There are formulas and instructions for choosing the dose of a human drug for use in experimental animal models (e.g., doi:10.1096/fj.07-9574LSF). Similarly, the question of choosing the dose of antioxidants can also be raised.

3. In describing the study's results, the author relies only on statistical analyses (more, less, the same) without mentioning specific obtained values. And according to the data in the images, we have unexpectedly significant changes in the examined parameters (e.g., a 250% increase, not a decrease (line 144) in the concentration of FSH in the vancomycin group). The hypothalamus-pituitary-endocrine glands axis is one of the body's most essential regulatory homeostatic mechanisms, so hormone concentrations change little in shorter intervals. That's why it seems suspicious that this kind of result is possible after only seven days of treatment, even though it appears to be too large (very toxic) doses of the drug for rats, which are confirmed by the histological analyses of the testicles.

4. The author does not compare his results with literature data, for example, whether there are any similar changes obtained with another antibiotic or a combination of the chosen antibiotic and other food antioxidants. Generally speaking, the entire text of the work is full of generalities and assumptions that cannot be directly linked to the specific subject of the research.

Author Response

Reviwer 2

In the submitted manuscript (medicina-2232959), the author examined the effect of a short administration of the antibiotic vancomycin on testicular function in rats, as well as the protective effect of the antioxidant resveratrol on worsened monitored parameters, such as the level of testosterone, follicle-stimulating hormone, luteinizing hormone in the blood, sperm count and mobility. Morphological changes in the testicles as a consequence of the treatment were also monitored. Under the applied conditions of the experiment, the author reports dramatic changes in the monitored parameters in the animals treated with the antibiotic, especially those in the blood, which are all normalized (i.e., they corresponded to the controls) if the antibiotic is added with an antioxidant.

My overall evaluation of the existing text is not favorable. Apart from the idea of the entire study, which is excellent, everything else, starting with the design of the study, the lack of details of the performed experiments, the description and commenting of the obtained findings, and their placement in the context of the literature in this field, is unfortunately not at the required level.

The paper's technical quality should be much better, avoiding unclear sentences, countless typing and editing errors, material mistakes, and unnecessary repetitions (for example, it is enough to say once in the summary of the work how long the treatments lasted), etc.

I would like to thank the reviewer for this comment. The abstract was revised.

The most critical essential remarks are the following:

  1. Duration of the treatment: why did the author decide that it only lasts seven days? Vancomycin is an antibiotic used in humans for severe pathophysiological conditions for longer than one week.
  2. Why did the author apply a daily dose of antibiotics of a large 200 mg (i.p.) to a body weight of only 200 g? How old were the experimental animals anyway? Adult ones are usually heavier (more than 300 g after three months of life). There are formulas and instructions for choosing the dose of a human drug for use in experimental animal models (e.g., doi:10.1096/fj.07-9574LSF). Similarly, the question of choosing the dose of antioxidants can also be raised.

I would like to thank the reviewer. The dose of resveratrol was selected based on several studies that used 20 mg/kg to show its ability as an antioxidant and anti-inflammatory effects [30-32]; the vancomycin dose was selected based ability to produce toxic effects on several organs, including testicular tissues [33-35]. 

These studies were used as guidance for this experiment regardless of its regular use in humans.

The animals were 7 weeks. The age was updated in the text.

  1. In describing the study's results, the author relies only on statistical analyses (more, less, the same) without mentioning specific obtained values. And according to the data in the images, we have unexpectedly significant changes in the examined parameters (e.g., a 250% increase, not a decrease (line 144) in the concentration of FSH in the vancomycin group). The hypothalamus-pituitary-endocrine glands axis is one of the body's most essential regulatory homeostatic mechanisms, so hormone concentrations change little in shorter intervals. That's why it seems suspicious that this kind of result is possible after only seven days of treatment, even though it appears to be too large (very toxic) doses of the drug for rats, which are confirmed by the histological analyses of the testicles.

I would like to thank the reviewer. The toxic effects of vancomycin on FSH levels have not been studied before. However, several studies have shown that FSH can change in 1 week to 10 days after toxic conditions. https://doi.org/10.1016/j.bbrep.2021.100999

https://journals.sagepub.com/doi/pdf/10.1177/0748233706071739?casa_token=KNJFOPwOR_8AAAAA:rPC-HJ0qBRP2zOt2AFv0qjdLHN4I-2gg6Mrcnx0Umujpi38fsvD46fUXrVpcy0zLByAQ2FsIEJJHhw

  1. The author does not compare his results with literature data, for example, whether there are any similar changes obtained with another antibiotic or a combination of the chosen antibiotic and other food antioxidants. Generally speaking, the entire text of the work is full of generalities and assumptions that cannot be directly linked to the specific subject of the research.

I would like to thank the reviewer; the discussion part was updated significantly

Round 2

Reviewer 2 Report

In the revised version of the manuscript, the author made a commendable effort to supplement the text with the necessary details and clarifications. Although there are reservations about the study's design and the way of presenting and describing the obtained experimental results, the paper is good enough for publication. All the best to the author in his further scientific work.